# Situating the Nonprofit Industrial Complex

Tyson Singh Kelsall [1,*], Jake Seaby Palmour [2], Rory Marck [3], A. J. Withers [4], Nicole Luongo [5], Kahlied Salem [6], Cassie Sutherland [6], Jasmine Veark [7], Lyana Patrick [1], Aaron Bailey [8], Jade Boyd [9], Q. Lawrence [6], Mathew Fleury [1], Alya Govorchin [1], Nathan Crompton [10], Chris Vance [11], Blake Edwards [6], Anmol Swaich [1], Amber Kelsall [12], Meenakshi Mannoe [13,14,15], Portia Larlee [16] and Jenn McDermid [17]

[1] Faculty of Health Sciences, Simon Fraser University, Burnaby, BC V5A 1S6, Canada
[2] School of Social Work, University of British Columbia, Vancouver, BC V6T 1Z2, Canada
[3] Community Scholars Program, Simon Fraser University, Burnaby, BC V5A 1S6, Canada
[4] Department of Gender, Sexuality, and Women's Studies, Simon Fraser University, Burnaby, BC V5A 1S6, Canada
[5] Independent Researcher, Vancouver, BC V6B 5K3, Canada
[6] Independent Researcher, Vancouver, BC V6A 1H4, Canada
[7] Independent Researcher, Vancouver, BC V6B 0R1, Canada
[8] Eastside Illicit Drinkers Group for Education, Vancouver Area Network of Drug Users, Vancouver, BC V6A 1P4, Canada
[9] Department of Medicine, Faculty of Medicine, University of British Columbia, Vancouver, BC V5Z 1M9, Canada
[10] Vancouver Area Network of Drug Users, Vancouver, BC V6A 1P4, Canada
[11] Political Science Graduate Program, York University, Toronto, ON M3J 1P3, Canada
[12] School of Nursing, University of British Columbia, Vancouver, BC V6C 1S4, Canada
[13] Pivot Legal Society, Vancouver, BC V6A 3E9, Canada
[14] Vancouver Prison Justice Day Committee, Vancouver, BC V5N 5W1, Canada
[15] Defund 604 Network, Vancouver, BC V5Y 1L7, Canada
[16] Independent Researcher, BT7 3GZ Belfast, Ireland
[17] Interdisciplinary Studies Program, University of British Columbia, Vancouver, BC V6T 1Z1, Canada
[*] Correspondence: tsa154@sfu.ca

**Abstract:** This article centers on the nonprofit landscape in Vancouver, Canada, a city that occupies the territories of the xʷməθkʷəy̓əm (Musqueam), sḵwx̱wú7mesh (Squamish), and səlilwətaɬ (Tsleil-Waututh) nations, which have never been ceded to the colonial occupation of Canada. Vancouver has a competitive nonprofit field, with an estimated 1600+ nonprofits operating within city limits. This descriptive review starts by defining what a nonprofit industrial complex (NPIC) is, then outlines an abbreviated history of the nonprofit sector on the aforementioned lands. The article then explores issues related to colonialism, anti-poor legislation, neoliberal governance, the fusing of the public and private sectors, and the bureaucratization of social movements and care work as mechanisms to uphold the status quo social order and organization of power. Focusing on under-examined issues related to the business imperatives of nonprofit organizations in the sectors of housing, health and social services, community policing, and research, this work challenges the positive default framing of nonprofits and charities. Instead, we contend that Vancouver's NPIC allows the government and the wealthy to shirk responsibility for deepening health and social inequities, while shaping nonprofits' revenue-generating objectives and weakening their accountability to the community.

**Keywords:** nonprofit industrial complex; nonprofits; tenancy rights; housing rights; carceral housing; settler colonial; abolitionist social work; abolition; critical social work; social policy

## 1. Introduction

A nonprofit industrial complex (NPIC) protects status quo iterations of state, socioeconomic structures, and power by managing dissent (Farnia 2008, p. 281), quelling community demands to address inequities and allowing the state to shield itself from accountability.

NPICs permit prevailing conditions and political frameworks without upheaval, overthrow, or, as addressed in this article, significant reform. Industrial complexes are understood as the outcome of business models that are interwoven into political and/or state-run systems (or, at least, services that arguably should be publicly provided) (Choudry and Kapoor 2013, p. 3). The origin of the term can be traced to former United States President Dwight Eisenhower. Eisenhower referenced the "military-industrial complex" in his outgoing address, naming the process by which the arms industry systemically seeks to drive appetite for war, regardless of societal benefit (Eisenhower 1961, para. 18; Gilmore 2007). He noted that as the arms industry expanded, so did its influence on the political system; Gilmore (2022) interprets this as a warning about the arms industry invading every part of society. The NPIC reflects a similar logic. In the wake of impactful 1960s social movements, state apparatuses started and/or returned to funding and relying on nonprofits and other private sector business models to manage issues forcefully brought to the forefront. Spade (2015, pp. 29–30) positions the shift from 1960s grassroots organizing to the rise of NPICs as a re-entrenchment of neoliberal hierarchies over horizontal structures, which produces labor practices that are indistinguishable from other business models. Nonprofits, or third parties that combine private and public funding, often evade ideal levels of transparency and democratic responsibility, varying by jurisdiction; we argue that nonprofits, like the arms industry, seek to increase their own growth regardless of the expense placed onto the communities they operate within.

This article applies a NPIC analysis to a specific geography by drawing on and synthesizing a number of non-exhaustive but interconnected themes within the City of Vancouver, British Columbia for the first time. Vancouver occupies the territories of the xʷməθkʷəy̓əm (Musqueam), sḵwx̱wú7mesh (Squamish), and səlilwətaɬ(Tsleil-Waututh) Peoples, land that has never been ceded to the state, and we look at the layered provincial and federal policies that impact Vancouver's governance. After a brief examination of Canadian social welfare history, this article situates the NPIC in Vancouver through the exploration of multiple sectors. We interrogate the implications for community accountability and power as well as the expansion of carcerality and examine how the NPIC shapes the bureaucratization of care work. Though the political right is feeding off of distaste toward broken bureaucracies, this article shows how NPICs and the political right mirror one another in their approaches toward managing social issues, particularly through the constant expansion of carceral systems and a focus on revenue generation for the private sector. Previous NPIC analysis has focused on the sectors of health, education/advocacy, and social services (Clément 2019; Farnia 2008; Gilmore 2007; Rodriguez 2007; Samimi 2010). This article reflects these issues and expands the topic to include an analysis of community policing and research centers. We write this during a drug toxicity crisis that continues to kill 200 people per month in BC as of April 2023, despite being declared a public health emergency seven years prior (Pawson 2023). A direct interrogation into why this crisis persists has not been met with urgency from the state or the nonprofits it funds as of July 2023, yet both public and private bureaucracies have expanded under the guise of mitigating the emergency. Some nonprofits have received millions of dollars dedicated to the response for initiatives tangential to the crisis (Dalton n.d.; Singh Kelsall 2022, 2023). Our review comprehensively explores Vancouver's NPIC, the localized relationships between nonprofit agencies and the organization of power, and the question of how the NPIC can limit change and cause harm. We conclude with the notion that future exploration should work to develop alternatives to the NPIC.

## 2. Abbreviated History of Social Welfare in Colonial Canada

The mix of public and private social welfare infrastructure that emerged as Britain and France aimed to create formal colonies on the land that Canada occupies is part and parcel of the aims of colonialism and the criminalization of difference. Canada has attempted to remove diverse Indigenous Peoples, nations, and communities from the lands it has worked to possess; this dispossession is required for Canada to underwrite its own "state formation and capital accumulation" (Coulthard 2020, p. 379). The establishment of disparate

Christian churches was at the forefront of colonial occupation; children were and remain the central targets of Canada's settler-colonial occupation and cultural genocide (Blackstock et al. 2004; Coulthard 2014, pp. 124–26; Luxen 2015; Milloy 1996, p. 18; Singh Kelsall et al. 2023; Young 2015). The funding structure of Canada's so-called Indian residential school system (IRSS), which operated from 1831 until 1996, functioned similarly to today's third-party nonprofits: government (public) provided the financial, policy, and land resources to establish these schools, and churches (private) oversaw their operation (Satzewich and Mahood 1995). Jeffery (2009) has described the roles of workers in these schools as being the "Canadian state's handmaidens" (p. 51). Million (2020) adds that "settler states are established through the violation of Indigenous reproduction; through the decimation of life-sustaining relations" (p. 395). The IRSS and the larger project of dispossession was upheld in part by Indian Agents, a police officer-like role occupied by a person who was hired as a government employee, whose appointment was often influenced by the churches (Satzewich and Mahood 1995). Displacement is at the foundation of what Canada is (Blue Sky et al. 2022), and private-public organizations have played a key role in this.

The settler-colonial project of Canada, which began as a state disguised as a merchant (Vaillant 2023), has always relied on and weaponized nonprofit and similar private sector organizations to control some communities and populations through social welfare activities, since its formation from the Hudson's Bay Company, including prior to confederation. By 1837, Ontario had developed charity-operated workhouses, institutions in which poor people were confined and forced to labor for food and shelter (Chapman and Withers 2019; Palmer and Heroux 2016). Other provinces, including Nova Scotia and New Brunswick, held pauper auctions until the mid-19th century, where poor people were auctioned off to the entity that bid the lowest to care for the poor (Clément 2019; Guest 1997). Nonprofit organizations operated relief agencies that surveilled the poor, requiring temperance and civility in exchange for meagre aid in many communities. By 1912, Charity Organization Societies (COS) had established itself in Toronto, Montréal, Winnipeg, and London. The COS, later morphing into United Way, was a bureaucratic organization dedicated to monitoring charity assistance to ensure poor people were administered the smallest surpluses possible, or none at all (Chapman and Withers 2019). Many of Canada's social welfare policies were based on the British Poor Law—punitive legislation that existed in name from 1834 to 1901, which continues to influence social policy in BC (Bergsma et al. 2023; Green et al. 2021; Guest 1997). There are similarities between these early public-private charitable organizations and nonprofits of the contemporary period: both produce status and a salary for those in upper-level positions; both use business structures that underbid each other and subject people living in poverty to surveillance, violence, and control. During the earlier period, white supremacy and social abandonment meant that Black, Chinese, Panjabi and Sikh, Indigenous (Chapman and Withers 2019), and likely other communities were excluded from any benefits of social welfare and were at times subject to coerced labor and institutionalization (Elson 2011). These groups engaged in community mutual aid to support one another, and only a small fraction of this aid took the form of a nonprofit (Chapman and Withers 2019).

Major socioeconomic events, including the Great Depression and World War II, unsettled the inertia of social welfare in Canada. There was a shift in the distribution of resources, particularly in favor of a dominant class of nondisabled settlers, who had created considerable wealth through white supremacist colonial processes (such as ownership over devalued, stolen land) (Coulthard 2014, p. 4; Starblanket and Hunt 2018). This disruption created an extensive policy window, pushed open by nationwide insurgent labor actions prior to, during, and after the first steps toward nationalized health and social programming (Huberman and Young 2002; Logan 1928). Preliminary stages of socialized services like employment and health insurance saw iterations after the release of the 1935 Royal Commission on Industrial Relations (Thompson 2015). The first attempt at health insurance was struck down by the Supreme Court of Canada in 1936 (Guest 1997; Supreme Court of Canada 1936), but the federal government continued to build a welfare state. The federal

government did slowly implement a version of socialized healthcare over the next 50 years, despite pushback from some sectors such as physicians (Martin et al. 2018; Santhanam 2020). This broad policy framework notably distinguishes the NPIC context in Canada from that in the US in terms of how the health sector is regulated and how competition is shaped. Between the 1930s and 60s, governments also continued to primarily fund nonprofits styled after Poor Law logics that were registered as charities. By the 1960s, both federal and provincial governments began funding broader nonprofit-type groups as a response to effective Indigenous rights, anti-racist, gender, and disability rights initiatives, and labor-oriented social movements. This included funding for nonprofits that were advocacy- and/or community service-based (Mulé and DeSantis 2017, p. 4). An abrupt turn to neoliberal governance in the 1970s and 80s would see a burgeoning of nonprofits—a process that continues in Vancouver.

## 3. Legislative Framework of BC Nonprofits in the 21st Century

In Vancouver, the provincial *Societies Act* (2015) regulates how provincial nonprofits are incorporated, operated, and dissolved. The *Societies Act* (2015) stipulates that all societies must have lawful intent and cannot conduct for-profit business ventures, with exceptions for activities supporting the society's mandate. Reasons to be considered a nonprofit vary widely and can be for the benefit of a few people (BC Government n.d.). To become incorporated, entities must draft a constitution, codify bylaws, and publicly identify contact information for their boards of directors. Societies are required to host annual general meetings prior to the end of each fiscal year to ensure that the operational direction is endorsed by its members (BC Government n.d.). Nonprofits must make the remuneration of their highest-paid employees public, and directors are expected to disclose and recuse themselves from conflicts of interest (Ullrich and Blackstock 2023).

Minor amendments to the Societies Act in 2023 were touted as mechanisms that "increase transparency" (Ullrich and Blackstock 2023, para. 1). However, nonprofits remain unobligated to release detailed accounting records, meeting minutes, or annual reports for public review, although funding bodies may request this information on an ad hoc basis (People's Law School 2023; *Societies Act* 2015). Public funding is a crucial revenue stream for nonprofits, yet standards for the release of this information mostly do not exist (Clément 2019). Additionally, service users have few means to remove board members who act against community direction, misuse funds, or preside over harmful practices (*Societies Act* 2015); nonprofits are likewise not subject to freedom of information policies. The structure of the act allows for nonprofits to be shielded from existing mechanisms of accountability.

## 4. Vancouver's Endless Nonprofit Landscape

The nonprofit landscape in Vancouver is complex and condensed, especially in the downtown eastside (DTES), a community characterized in part as a neighborhood where residents build informal networks of care in the face of organized abandonment (i.e., Askey 2023; Wells 2023). The neighborhood, which stretches along the city's northeastern coastline, is colloquially referred to as Canada's "poorest off-reserve postal code". The region is named k'emk'emeláy in the language of the Sḵwx̱wú7mesh Úxwumixw. Demographically, the DTES is unique from the rest of the city in its multi-racial composition; for example, the previous literature shows that 70% of the Indigenous population in the city reside in the community (Benoit et al. 2003). Though the boundaries of the DTES shift, one analysis shows that 43.9% of residents spoke English as a first language, compared to 40.3% with either Mandarin or Cantonese as a first language (Vancouver Coastal Health 2013, p. 72); comparatively, 87% of the city's total population spoke English as a first language (Statistics Canada 2011). The DTES is also home to many people who are precariously housed or are living outside; in 2020, one report showed that Black people are 3.7 times more likely to experience homelessness, and Latin American and Arabic people also experienced higher-than-average rates of homelessness in the city (BC Non-Profit Housing Association 2020).

The DTES is a cultural hub for many non-residents, with deep histories embedded in the Indigenous, Black, Japanese, and Chinese diasporas (Assam 2023; City of Vancouver 2013).

A 2020 municipal government document estimates that 1660 nonprofits operate within city limits, many of which have multiple sites. (This total does not include those in bordering urban hubs, like Surrey or Burnaby.) Across BC, the nonprofit sector generates approximately CAD 6.4 billion in gross domestic product (City of Vancouver 2020). The City of Vancouver (2020) outlines nonprofits as playing a "vital role in the health of communities" (p. 5). However, the negative impacts and opportunity cost produced by the sector remain under-examined, as do the outcomes of having such a high number of competitive nonprofit organizations. The limits of nonprofits to pursue improved material and social conditions for the communities they claim to serve also warrant critical interrogation.

Limitations of the NPIC in Vancouver were brought into focus in 2023, when the governing civic party, A Better City (ABC), stated that all city grantees would be required to "to communicate to, about, and with city officials in a respectful manner" (Fumano 2023, para. 9). This was a reinforcement of respectability politics/civility into the administration of public grant money. Only one service provision nonprofit reacted with public critique (see: Koncan and Burrows 2023), an example of nonprofits adhering to a threat from a funding body. This decision came shortly after ABC had defunded a small CAD 7500-per-year art program hosted by an advocacy organization called the Vancouver Area Network of Drug Users (Ayers and Nassar 2023). There is evidence that nonprofits that do not receive government funds are more able to conduct discursive and direct service position activities in opposition to government (Withers 2021). These examples show how fear of funding loss can silence nonprofits.

The interplay between nonprofits, transparency, and community justice was brought to the news cycle for a second time in 2023. A forensic audit implicated CEO Shayne Ramsay of BC Housing, the provincial government housing arm, in a conflict of interest over numerous issues involving his spouse, Janice Abbott, then CEO of Atira, a large nonprofit housing provider (Ernst and Young 2023). The province's governing New Democratic Party ignored systems-level questions, at least publicly, and instead framed this as a case of one bad apple (Depner 2023).

*4.1. Housing*

In Vancouver, a small number of nonprofits receive the majority of public resources dedicated to housing. The sector is characterized by unsafe infrastructure (St. Denis 2023c; Judd and Dao 2023; Robinson and Judd 2023), limited tenancy rights (St. Denis 2023b, see also Appendices A and B), and surveillance (Boyd et al. 2016). The term "supportive" housing is used here only to reflect sociolegal language. Tenants who live in "supportive," "transitional," and/or "temporary" residences all formally have fewer rights than those living in the private sector in Vancouver (Residential Tenancy Act 2023, Section 4). Supportive and temporary housing programs are an extension of the local carceral continuum tied to police collusion, punitive sub-tenancy rights, and surveillance (Atira Property Management v. Richardson 2015; Boyd et al. 2016; Fleming et al. 2019; PHS Community Services [PHS] v. Swait 2018). The DTES, where much of this housing stock is located, shares characteristics with and a high concentration of direct connections to the carceral system. The DTES has an intensive presence of parole and probation infrastructure (Crier et al. 2021), as well as a high concentration of police who share a plurality of relationships with healthcare services; and supportive housing is central to this machinery. "Spatial conditions of release" from incarceration to highly criminalized and regulated spaces, such as the DTES and specifically supportive housing sites, means that individuals, once criminalized, are under intensive surveillance post-release, and they have a myriad of behavioral conditions tied to their discharge, including red zones (area restrictions) (Sylvestre et al. 2020, p. 140; Sylvestre et al. 2017). In a housing crisis context, if one is enduring life at a supportive site, there are few pathways to alternative housing. The cost of the cheapest private one-bedroom apartment in Vancouver has eclipsed the rate of a

full monthly disability income assistance payment (Manson et al. 2022). This can trap a person within a few city blocks indefinitely (Roe 2010). BC Housing also lists a number of exclusion criteria to apply for limited non-supportive housing options; these include "anti-social" and "nuisance" behavior, drug use, and "begging" as examples (BC Housing 2023, sct. e). Boyd et al. (2016) have described supportive housing buildings as "sites of social control" (p. 73). Carceral aspects of these sites range from video surveillance of communal spaces to integrating police directly into their operation. Police are commonly included on advisory committees and in tenant selection processes (i.e., Coast Mental Health 2019; Vancouver Police Department 2020). They are often invited to sit on boards of directors (i.e., Coast Mental Health 2023) and have regular meetings with housing and shelter providers and outreach-based services (Vancouver Police Department 2020). This includes sometimes being informed of residents who are "barred," a temporary ban based on concerns upheld by individual building criteria and staff discretion. Supportive housing staff and management frequently collude with the Vancouver Police Department (VPD) to criminalize residents (Boyd et al. 2016; Vancouver Police Department 2020). In addition, if a tenant is receiving social assistance and experiences incarceration for more than three months, through this collusion or otherwise, the BC government halts shelter payments, putting the tenant at risk of eviction (Bergsma et al. 2023). By partnering with police and other appendages of the surveillance state, supportive housing staff, many of whom live at the intersections of anti-Indigenous sentiment, racial capitalism, gender discrimination, disablism, and/or criminalized drug use, may experience normalized psycho-affective attachments to this spatial-carceral colonial violence (Coulthard 2014, p. 17); this prejudice is defined in part by a colonial subject abandoning their own cultural ties to take on those of the colonizer (Fanon 1963, p. 219), sometimes referred to as "colonization of the mind" (Puri 2011, p. 7). This coerced complicity, generated by the NPIC, can assuage those with policy power within this structure to see supportive housing-related harms as regrettable but natural parts of working within and against colonial injustice, rather than as abhorrent and calculated forms of colonial violence (Coulthard 2014). Program agreements, enforced in part by staff and management, subject tenants to additional forms of control, such as strict guest policies and commitments to being a good neighbor, a widely interpretable descriptor (Fleming et al. 2019; see Appendices A and B for examples). These agreements rely partly on nonprofit housing sites being labeled as therapeutic (see: Appendix B), not dissimilar to the repression of prisoners' rights by naming prison labor, health, and shelter "rehabilitation" (Jackson et al. 2022, p. 28). Program agreements were successfully challenged at the level of the BC Supreme Court in one case (PHS Community Services [PHS] v. Swait 2018), as was the requirement for visiting guests to show identification (ID) to enter in another (Atira Property Management v. Richardson 2015). However, program agreements, including ID requirements, remain commonplace in supportive housing buildings. Both red zones and ID requirements, weaponized in Vancouver's community with the highest proportion of Indigenous Peoples, replicate logics of the pass system once upheld by Indian Agents and police alike. This system forced Indigenous People to seek permission from the settler state to leave a reserve or else risk apprehension, a policy that existed to some degree from 1893 until it was abolished in 1941 (Smith 2016). This collection of policies likewise reflects broader apparatuses that restrict the movement of migrants, racialized people, those with disabilities, and/or those living in poverty (Walia 2013). Nonprofits create their own versions of these agreements, meaning the network of supportive housing has inconsistent rights and regulation. A PHS program agreement from 2023 can be found in Appendix B, including a clause that coerces tenants into allowing PHS to collaborate with external health and social service agencies, as well as a commitment that rent paid via social assistance goes directly to a landlord, limiting tenant power to control or withhold rent money. Between 2021 and 2023, government hotel takeovers and other housing projects in the city were mostly transitional, rather than supportive, a clear decoupling from normative tenancy rights via the RTB and the residential tenancy act that defines it (Appendix A; Residential Tenancy Act 2023). Transitional/temporary housing, including abstinence-based and

drug treatment housing, comes with no formal tenancy rights, and people can be evicted without notice. Supportive housing remains in a legal grey area (BC Government News 2022; Residential Tenancy Act 2023; St. Denis 2023b). Nonprofit housing in Vancouver is defined by temporary accommodations as the dominant form of subsidized housing, whereas permanent, independent, family, and multi-unit housing remains scarce (Manson et al. 2022). Temporary and single-occupancy sites generally come with restrictions on partners residing together (Manson and Fast 2023; Manson et al. 2022). Along with strict guest policies, this separates families living in poverty and creates a barrier to supportive housing among tenants raising children, arguably a structure aimed at the erasure of certain demographics. Tenants, whom the industry is built upon, experience precarity, unstable shelter, and largely inescapable poverty, and nonprofits economically benefit from keeping people stuck within the cycle of the system from which they generate revenue.

All levels of government, including BC Housing, have done little to hold nonprofits and/or the landowners who subcontract them accountable in keeping buildings safe. This includes limited use of expropriation, enforcement of fines, or other measures (St. Denis 2020). A number of these buildings lack adequate or bare-minimum accessibility design, such as functioning elevators or wheelchair-oriented infrastructure and/or sufficient space (Peters 2019, p. 10; Vescara and St. Denis 2023). Tenant safety risks have been highlighted in numerous reports of malfeasance at nonprofit housing sites (St. Denis 2023c), including overdose deaths (Lupick 2020), assaults, and murders (Little 2020). In addition to a lack of safety, tenants report that there is negligible recourse via internal complaints (Boyd et al. 2016; Fleming et al. 2019; Fleming et al. 2023). Financial resources that could ameliorate tenants' unsafe and unsustainable living conditions flow through several layers of opaque bureaucracy, and a sizable portion of housing funding is lost to government and nonprofit executive salaries and administrative expenses (Ernst and Young 2023; St. Denis 2023a). There are likewise added costs of having several streams or outsourcing of all appraisers, maintenance teams, etc., compared to having one centralized hub. BC Housing does not issue requests for proposals regularly, nor are processes for awarded contracts transparent to the public (Bula 2022; Ernst and Young 2023, p. 15). State response to Vancouver's growing housing crisis has been to increase the reliance on the NPIC to provide shelter, and the private market is now out of reach for the majority of BC residents (Meissner 2022). This dynamic generates revenue for organizations, while tenants experience unsafe infrastructure, few rights, and the constant presence of carceral surveillance. These conditions push tenants to street-level homelessness, which has been shown to be a negative social determinant of health (Cowan et al. 2007; Public Health Ontario 2019; Singh Kelsall and Mannoe 2022).

*4.2. Health*

Subpar housing and other forms of violence produced by poverty have created a demand for a healthcare nonprofit industry. In Vancouver, nonprofit community health centers (NP CHC) exist at this intersection. NP CHCs rely on precarious funding that comes with adherence to government and private sector/donor objectives. This is in spite of NP CHCs in Canada typically emerging "as a result of social activism, to meet…the unmet or poorly met needs of marginalized populations" (Lavoie et al. 2018, p. 2). Some researchers argue that NP CHCs are an improvement to Eurocentric models, as the latter tend to perpetuate white supremacy by meting harm on non-white communities (Allan and Smylie 2015; Hantke 2022; Lavallee and Harding 2022; Stelkia 2020); proponents say the workforce of NP CHCs more frequently share community and kinship ties, ethnicities, and other sociocultural positionalities with service users and can therefore provide care that is anti-racist, relational, and justice-oriented (Baskin 2022; Greenberg et al. 2018; Lavoie et al. 2018; Linklater 2014; Oleman 2022). It has been argued that competition between nonprofits fosters innovative solutions for under-served communities (Lavoie et al. 2018). These arguments position NP CHCs as better able to anticipate and respond to emergent local crises, to promote social justice, and to educate public health officials and other bureaucrats on appropriate responses (Lavoie et al. 2018, p. 9) However, notwithstanding the emancipatory

potential of the service provision "for us by us," these arguments may be chimerical. NP CHCs satisfy the reductive imperatives of "new public management," as their undercompensated labor force fills a void left by an inadequate and further shrinking welfare state (Cunningham et al. 2014; Moffatt et al. 2017). The question as to whether competition can breed social innovation in some situations should not override the problematization that as a driver for healthcare services, it compels the corporatization of healthcare, a process that does not always align with the health and wellness of a community (Sanders 2015). NP CHCs can become hamstrung by the mandates and lenses of their neoliberal funding masters (Choudry and Kapoor 2013, pp. 13–14; Lavoie et al. 2018; Moffatt et al. 2017). It is possible that NP CHCs can *anticipate* impending health crises and emergent community needs, but their ability to *respond* to these can be impeded by stringent, short-term performance indicators, the specificity of which often fail to capture the true breadth of services required by users (Lavoie et al. 2018; Moffatt et al. 2017). Further, enumerating performance indicators requires significant labor allocation, and many NP CHCs struggle with staff retention and recruitment for reasons of wage and administrative workload because of their precarious funding structure (Cunningham et al. 2014; Lavoie et al. 2018; Patrick 2019). Chronic labor shortages then impact the ability of NP CHCs to meet performance targets, and an ever-changing, under-remunerated, and increasingly technocratic workforce can contribute to diminished quality of care for recipients (Cunningham et al. 2014; Lavoie et al. 2018; Moffatt et al. 2017; Yee 2017). These factors intersect with the proclivity for Western health providers to refer community members with needs considered complex by their own intensely biomedical assessments to NP CHCs. This process places pressure on strained nonprofits, whose practices are restricted in flexibility by the very agencies administering referrals with one hand and funding conditions with the other. This dynamic exonerates clinicians in traditional health settings and the public health system itself when it/they fail to provide equitable, anti-oppressive services (Lavoie et al. 2018; Schill and Caxaj 2019). Million (2020) writes that "Liberal humanitarian tropes are wearied at this point", and that these "relations sever and kill what makes us possible to be cohesive peoples who act from. . .the practices that produce life for ourselves and all the entities that we are reciprocally responsible to" (p. 402).

Reliance on funds allocated by the state and the wealthiest classes limits NP CHCs in their orientation toward advocacy. The competition imperative that pits agencies vying for public dollars against one another can "undermine relationships among those who must collaborate to ensure that community members can access a broad range of services that fit their needs" (Lavoie et al. 2018, p. 7; Tang and Wang 2022). The fiduciary power of these settler-colonial public agencies interferes with "health sovereignty" (Million 2020, p. 398), as it constrains autonomy over culturally specific health interventions (Joseph 2023). Million (2020) contends that Western medical treatment "does not extend the supports necessary for lives still lived under siege" (pp. 398–99). NP CHCs could be utilized to influence public health, but they are typically positioned as health structures that are "lesser than" (Lavoie et al. 2018, p. 10). Knowledge sharing between the two is obscured through power and effectively does not exist (Lavoie et al. 2018).

### 4.3. Research in Communities

NPIC-based research and data collection can also be extractive activities (Gaudry 2011, p. 113). Gaudry (2011) asserts that researchers are frequently removing "deeply meaningful information" from community contexts (p. 113) and adds that "few researchers are willing to acknowledge a major responsibility to the communities that they serve" (p. 113). A group of DTES residents and researchers published a set of principles in response to this lack of reciprocity and a roadmap for sharing in the benefits of data and evidence (Neufeld et al. 2019). Reciprocity is not the sole tension in research when it is extracted from communities that experience marginalization by the state or another power. A related phenomena is "damage-centered research" (Tuck 2009, p. 413). Damage-centered research can be problematic beyond the NPIC, but it reflects NPIC logic when we examine

who benefits and who bears the cost. Tuck (2009) defines damage-centered research, in part, as methods that involve people from a community being studied but relegate their involvement to speaking from a place of pain, suffering, or tragic circumstance (p. 415). Tuck (2009) contends that this form of research sees systemic issues "submerged" (p. 415), if even acknowledged, beneath interventions that focus on a single, rapidly measurable hypothesis, which can obscure the drivers of issues being examined. For instance, another study on buprenorphine/naloxone or methadone—long-acting opioids that benefit a small portion of users—might elevate these medications as substance use treatment, while also reinforcing the social license of the state to delay addressing the much more significant, well-founded systemic harms associated with drug prohibition (Ray et al. 2023; Werb et al. 2008). Tuck (2009) uses the example of no-child-left behind policies similarly propping up inaction toward the root cause, namely, state-manufactured poverty, in neighborhoods where this program is applied. These instances focus on a symptom of a social issue rather than its cause. A similar logic can be applied to the myriad of DTES research nonprofits that are equity- and justice-focused in name but that remove focus from broader systems of power and place it instead onto micro-interventions within the status quo; moreover, researchers themselves have traditionally been the primary beneficiaries of these projects through career advancement, wages, and policy influence, although the increasing precarity of academic work has tilted this dynamic (Rose 2020).

　　Criteria of what should be studied in order to acquire funding means that researchers are limited in their own discretionary parameters of inquiry. In BC, there are examples of high-ranking officials, such as then-Attorney General David Eby, influencing publicly funded research through his staff, with preconceived conclusions about the existence of "prolific offenders" (Salter 2022, p. 10), a label that does not have criteria or a definition (Butler and LePard 2022, p. 89) but is politically malleable. Likewise, the City of Vancouver routinely awards research nonprofits with funding to conduct equity-based evaluations that are related to social or environmental issues, especially within the DTES. Although equity is a valuable lens, research funds granted from the municipal government could be allocated directly to the urgently needed public infrastructure. Though these are relatively small grants (see: Blyth-Gerszak 2023), the phenomenon illustrates how governments invest substantive resources into research projects to delay the implementation of tangible solutions that would address inequalities, paralleling the objectives and the class of beneficiaries typical of the NPIC. The politics of funding mechanisms can see a single nonprofit simultaneously produce evidence that would solve societal issues, while being financed to evaluate and/or build programs that almost certainly will not. For example, a nonprofit may focus on expanding access to regulated drugs that would reduce overdose death rates, while also being funded to uphold a small but expensive inpatient drug treatment program that evidence suggests will not have a positive impact on drug toxicity death rates (Hayes 2023) but is rather likely to increase the risk of overdose for clients in a toxic drug crisis context (Morgan et al. 2020; Singh Kelsall et al. 2023). The latter paradigm entrenches neoliberal socioeconomic systems by placing individualized blame (e.g., adherence to abstinence) rather than addressing systemic-level causes (i.e., the devastating impacts of prohibition policies) (Patrick 2019)—something wealthy philanthropists and politicians are willing to fund.

　　As research-based universities face decades of neoliberalization, funding is increasingly administered via competitive streams. Grant writing, judging, and then contending with wasted time if an application is unsuccessful all come with the lost opportunity cost of community-oriented action (Dresler 2022). In BC, there are extremely limited sources of non-competitive funding that do not double as internal evaluations or come with predetermined objectives, and aside from administrative opportunity cost, this fosters competition in research rather than collaboration. The outcome of this competition means that proximity to wealth and time dictate who can attempt to compete. This often means that communities are left out of guiding the research that they will be directly impacted by, and that funding bodies have considerable power over the production of evidence, as well as who is paid and included as members of a community advising committee and/or leading projects.

*4.4. Policing*

The NPIC also shapes the contours of Vancouver's policing sector. There are least eleven nonprofit, civilian-staffed and administered Community Policing Centres (CPC) in the city, five of which are located in close proximity to the DTES (VPD 2022). The VPD (2022) reports that CPCs engage in "patrols, road safety programs, ethnic and cultural education programs, senior safety programs, and victim services" (p. 37).

This additional layer of surveillance and enforcement of normative behavior exists in spite of the DTES as a place of refuge, resistance, and solidarity for many who are targeted by the process of colonialism, including policing (Boyd et al. 2016; Martin and Walia 2019). The neighborhood has a relatively high Indigenous and racialized demographic and is frequently portrayed by media and policymakers "as a bounded space of immorality, vice, and disorder threatening the presumed contrasting order" of the city (Boyd and Boyd 2014, p. 315). This framing legitimizes the enactment of perpetual and inordinately high levels of police violence upon residents (Carmichael and Kent 2015). VPD communication briefs with perfunctory statements such as "community matters" (see: VPD 2022, para. 1) are public relations strategies that fail to obscure the reality that police shootings, killings, and "intimidation and cruelty are mainstays of police presence in the neighbourhood" (Pivot Legal Society 2020, para. 2; Griffiths 2022; Hayashi et al. 2023; Spring Magazine 2022).

Nonprofit organizations have worked to legitimize and stabilize policing in the face of critique and calls for defunding. The 2022 Special Committee Report on reforming BC's Police Act centered on an acknowledgement that trust in policing institutions continues to wane within Indigenous and racialized communities (Special Committee on Reforming the Police Act 2022). To help restore Indigenous Peoples' trust, the report urges the province to "increase coordination and integration across police, health, mental health, and social services" (p. 79) and to establish "tiered policing models," which would replace police with a civilian-led response (p. 8). These adjustments, likely to belie an enhanced role for CPCs, may seem favorable to those who advocate for defunding and ultimately abolishing the police, but CPCs have accompanied rather than replaced increased funding for the VPD (Jang 2022; Mannoe 2023b). Drawing on Winnipeg, Treaty One Territory, Dobchuk-Land (2017) argues that Indigenous-led CPCs can empower the settler state by integrating "crime prevention and suppression [that] has been more a project of attempting to 'manage' urban Indigenous people than serve their interests" (2017, p. 405). Dobchuk-Land (2017) contends that the model positions the violence and inequities of colonialism as historical by shifting blame back onto Indigenous peoples. CPCs do not ameliorate material conditions or decrease inequities but can shift responsibility away from settler colonial police forces for carceral harm (Stelkia 2020), all while the state constructs "itself as a benevolent responder to Indigenous communities in the present" (Dobchuk-Land 2017, p. 405). Fundamentally, CPCs remain enforcers of colonial law that are strongly tied to traditional police forces.

In Vancouver, CPCs are generally registered as nonprofits and/or charities. CPCs are overwhelmingly government-funded, and like other nonprofits, there is little in place to ensure that they pursue community betterment. CPC municipal funding flows through the VPD, and they are promoted by the VPD as one reason for budget expansion (Jang 2022). In addition to CPCs, the VPD and Vancouver Police Foundation have a number of arm's-length programs (Vancouver Police Foundation 2023), many of which are dedicated to promoting police and often convey false or misleading information. This includes the Odd Squad Drug Awareness program, a campaign used to spread fear about drugs to youth while ignoring harm caused by the police as enforcers of prohibition and drug seizures (Hayashi et al. 2023; Ray et al. 2023). The Vancouver Police Foundation (VPF) is a registered charity that receives millions in annual revenue (Canada Revenue Agency 2022). In 2021, the VPF reportedly began administering a 5-year grant from a real estate developer, whose donation conditions included financing the CPCs located in closest proximity to the DTES (Hager 2021). The sociopolitical framing of residents of the DTES as dangerous is dominant in the media, validating the surveillance of "visible" poverty, fueled by the VPD positioning

themselves as experts on topics of which they are not, such as mental health and crisis response (Boyd and Kerr 2016; Singh Kelsall and Mannoe 2022).

## 5. Nonprofits, Power, and Community Accountability

Consistent with neoliberal values, nonprofit outputs, goals, and functioning are closely tied to funding structures and relationships between funding bodies (Alexander and Fernandez 2020; Evans et al. 2005). Included in this are local, provincial, and federal governments, who generally account for roughly 60–80% of social and human service nonprofit funding in Canada, as well as the corporate or ownership classes who regularly funnel money through registered nonprofit and charitable organizations as a way to receive significant tax breaks (Evans et al. 2005; INCITE 2007; Samimi 2010). Other scholars, such as Tompkins-Stange (2016), have noted how large funders have become "strategic: in their funding agendas, interweaving themselves into progressive-seeming initiatives, including those that are presented as education-based, but ultimately shaping these projects to adapt to processes and norms from which they will benefit. The implications of these funding structures and relationships with the state and ownership class are significant, with the use of funding contracts and strict administrative accountability systems being implemented to solidify funder control of programming and organizational objectives (Evans et al. 2005). Critically, within this neoliberal context, nonprofits are often required to demonstrate organizational legitimacy to the state and the wealthy in order to access funding, which leads to the prioritization of accountability toward those bodies rather than the community and fosters a reliance on rigid, results-based performance measurements that are consistent with the profit-maximizing imperatives of extractive capitalism (Alexander and Fernandez 2020; Samimi 2010). As a consequence, there are observable gaps between community-led objectives and the practices of nonprofits, which have moved toward direct service delivery and away from advocacy and community accountability (Alexander and Fernandez 2020). The ability of nonprofit organizations to support meaningful community participation is therefore diminished, at the expense of those most impacted (Alexander and Fernandez 2020; Evans et al. 2005).

The reliance on the wealthy and political class of funding bodies produces environments where principles of "civility" within social justice movements are upheld. Presumptions about civility are inextricably tied to whiteness and white supremacy, whereby expressions of assertiveness and frustration around addressing inequities (or even expressive joy or celebration) are treated punitively and coded as problematic. Nonprofits established by dominant economic and political classes of elites have played an instrumental role in suppressing grassroots organizing efforts, which are often labeled as unnecessarily aggressive or unruly (Kherbaoui and Aronson 2021; Spade 2015). INCITE (2007) utilizes the term "buffer zone" to describe how the dominant classes "prevent people at the bottom of the pyramid from organizing to maintain the power, the control, and, most important, the wealth that they have accumulated" (p. 134). The buffer zone involves taking care of people just enough to keep hope alive to access adequate support, while at the same time providing a way for the beneficiaries of the system to maintain supremacy. By maintaining control through nonprofits, those in power are able to reduce the possibility of effective attempts to dismantle the white supremacist, capitalist, and patriarchal social order and to reinforce the legitimacy of state power (Alexander and Fernandez 2020; Kherbaoui and Aronson 2021). Further, for white leadership and funders, the NPIC allows those in power to channel white guilt into charity-oriented solutions that disproportionately fund white-led organizations that neglect to address structural racism, colonialism, and related structural inequities (Kherbaoui and Aronson 2021; Spade 2015). Incentives such as substantial tax reductions for the ultra-wealthy who contribute monetarily to nonprofits decrease incentives for the state to levy fair and rigorous taxation on the richest class (Duquette 2019). Emerging research shows that these subsidies are designed to benefit the wealthy by allowing them to maintain their immense fortune while appearing philanthropic (Duquette 2019). These tax deductions produce a class stratification through the influence of nonprofit organizations.

As outlined above, both decentralization and the fusing of public and private sectors have resulted in nonprofits taking on a significant role in service delivery in Vancouver. Neoliberal momentum toward a return to funding private sector solutions as the basis for public welfare has become a route to dismantling the role of the state in welfare provision, while suppressing community efforts to meaningfully improve people's living conditions (Evans et al. 2005). This move is exemplified in research by Black and Seto (2020), who highlight that social assistance recipients across BC have become structurally dependent on food banks and other charities to meet their basic needs, as welfare and disability rates remain at least CAD 10,000/year below the national poverty line. In 2022, 55.8% of food bank members in BC indicated that government assistance was their primary source of income (Grochowski and Crawford 2022). This is despite decades of nonprofit revenue and service growth in the city.

## 6. The Bureaucratization of Social Work and Care Work

Nonprofit sector tensions between secure funding and material equity likewise play out in service provision. Community service workers and those in related positions, who are themselves structurally marginalized and contend with precarity, may be coerced into immoral practices that include: upholding surveillance (Michaud et al. 2023); working within punitive and carceral policy frameworks (Bergsma et al. 2023; Boyd et al. 2016); and being coerced into additional layers of exploitation out of moral guilt, solidarity, and/or care for community (Mamdani et al. 2021; Patrick 2019, p. 174; Reynolds 2011). These layers can include withstanding a pervasive culture of white supremacy within a nonprofit (Anakwudwabisayquay et al. 2023; Asey 2022; Badwall 2016; Lavallee and Harding 2022). As funding is typically linked to institutions that benefit from the status quo, and which systematically reflect hegemonic whiteness, workers who are Indigenous, Black, or otherwise racialized may suffer direct psychological and material consequences from experiencing the contradictory desires to both serve their communities and contribute to their exploitation. All of this contributes to community service workers in BC facing extremely high levels of psychological injury compared to other fields, such as policing and paramedicine (Mental Health Commission of Canada 2021; Vescara 2023). As previously described, nonprofits tend to adhere to the current political system or else risk financial stability (Samimi 2010). For workers, this context may mean providing services within a depoliticized mandate. There are examples of workers risking speaking out on this type of organizational silence as complicity in times of state violence (see: Vancouver Coastal Health Employees 2023). This neutrality can rupture the therapeutic relationship between community-oriented workers and service users.

A pronounced NPIC produces what Graeber calls "total bureaucratization" (2015, p. 18). Graeber argues that this phenomenon occurs when the private and public become nearly indistinguishable. Healy (2001) has outlined shifts in social work practice toward managerialism. Healy (2001) argues that managerialism, or the growing reliance on administrative apparatuses, reduces worker ability to apply discretionary strategies born of critical (and, we argue, abolitionist and anti-colonial) theory. This shift sees community service labor become a negotiation, rather than an offering of material support or improved living conditions. Graeber (2015) posits that "deregulation" is a socially coded description of regulatory reforms, not necessarily less of them. Graeber argues that deregulation-oriented policy produces regulation that is more extractive and generates further bureaucracy (i.e., bureaucracy that focuses on surveillance, rather than support). These new layers tend to keep resources away from those who already have the fewest. Graeber (2015) labels this the "Iron Law of Liberalism," writing that any initiative to "promote market forces will have the ultimate effect of increasing the total number of regulations, the total amount of paperwork, and the total number of bureaucrats" (p. 9). This technologization and corporatization of social services—funded by government (public) and run by nonprofits (private)—sees workers negotiate and keep resources away from service users, thereby replicating the carceral logic of the 18th and 19th century Poor Law frameworks.

### 7. Limits of Analysis: The System Is Not Broken, It Was Built This Way

This article focuses on one NPIC in a specific sociopolitical context and city. Furthermore, state-run services in Vancouver have likewise been linked to issues of white supremacy (Birk 2018; Mannoe 2023a), elements of colonialism (Lavoie et al. 2018; Mannoe 2023a), disablism (Mannoe 2023a), and sexism and classism (Strega et al. 2002). Models like community-run cooperatives (Simmons and Birchall 2008; Thunder and Intertas 2020; BC Government 2023), mutual aid groups (Wells 2023), community-based land trusts (i.e., Assam 2023), participatory action research (Neufeld et al. 2019), and interventions into harm and violence (Barrie 2020; Howe 2018; Kim 2010) have shown promise. Significant change to electoral-political systems, nonprofit regulation, or legislative frameworks that strengthen community control over services by creating a new model may also be part of a solution. With or without challenges posed by an NPIC, issues related to austerity and resource allocation would persist.

### 8. Conclusions

This article contests the popularly perceived relation between nonprofits, charities, and the communities in which they function as inherently positive or neutral and argues that even if the framework can be utilized as a tool for change in some cases, Vancouver's NPIC may have a net negative impact. Though NPIC analysis exists in other contexts, this theoretical article is the first to explore it comprehensively within Vancouver, despite the concentration of nonprofit organizations in the city and the notable violence and harm produced by the NPIC that we have highlighted throughout this review. Nonprofits do not always provide equitable services and could be undercutting the potential for societal change by upholding the status quo social order and organization of power. We contend that in Vancouver, the NPIC has become a vessel for structures of neoliberal power, settler colonialism, racial capitalism, disablism, and carcerality to entrench themselves in the provision of human services, including through police budget expansion. The NPIC allows governments and politicians, foundations, and wealthy individuals to shirk responsibility by creating a facade of community-led initiatives that are underfunded and given inflexible mandates. Like any other industry, nonprofits in Vancouver seek to increase their own revenue growth, sometimes by producing further harm to and restricting access to resources from the populations from which they already extract revenue, in order to ensure the cycle continues. This organization of power creates a spectrum of further oppression for service users and/or workers who experience societal marginalization while working for nonprofits that refuse or neglect to respond to community needs based on fear of funding loss. Moreover, funding conditions are often dictated by the beneficiaries of these power structures, with little input from the people enduring ongoing oppression, which complicates the relationship between nonprofits and community-defined justice. Comprehensive resolutions and recommendations are valuable issues that future research should explore. Though these are beyond the scope of this review, we contend that all levels of governance that influence the City of Vancouver should consider alternative political futures to the NPIC.

**Author Contributions:** Conceptualization: T.S.K., J.S.P., R.M., K.S., C.S., B.E., M.M., J.M.; writing-original draft preparation: T.S.K., J.S.P., R.M., K.S., J.M.; writing—review and editing: T.S.K., J.S.P., C.S., A.J.W., N.L., M.F., J.V., L.P., A.B., J.B., Q.L., A.G., N.C., C.V., B.E., A.S., A.K., P.L., J.M.; validation: A.J.W., N.L., M.F., K.S., C.S., J.V., L.P., A.B., J.B., Q.L., A.G., N.C., C.V, B.E., A.S., A.K., M.M., P.L.; project coordination: T.S.K., J.S.P., K.S., M.F., J.M.; formation of methods: T.S.K., J.S.P., A.J.W., N.L., J.M. All authors have read and agreed to the published version of the manuscript.

**Funding:** The APC was funded by Simon Fraser University's Open Access fund; no other funding related to this review.

**Institutional Review Board Statement:** Not applicable.

**Informed Consent Statement:** Not applicable.

**Data Availability Statement:** There was no primary data involved with this review.

**Acknowledgments:** With gratitude to Kanna Hayashi for providing feedback on an earlier version of this paper, and to Jess Palmour, Tara Myketiak, Navi Dasanjh, Stephanie A., and D.S. for vital conversations. In solidarity with people impacted by this violent system, those who have passed, and those who grieve.

**Conflicts of Interest:** The authors declare no conflict of interest.

## Appendix A

**SUPPORTIVE HOUSING AGREEMENT**

C. **Background:**
This Supportive Housing Community provides Support Services to assist the Participant in addressing and enhancing life skills, restoring the ability to maintain healthy, independent lives and eventually maintain a productive independent residency. The Specific Support Services provided will be determined in consultation with the Participant and staff. The Participant agrees to utilize the Support Services offered, which include, but are not limited to:

    i. Directly assisting with room de-cluttering and/or normal cleaning and maintenance;
    ii. Individual or group support services such as life skills, community information, harm reduction practice, social and recreational programs;
    iii. Connecting the Participant to community supports and services such as education, employment, health and life skills and dependent residential tenancy opportunities when appropriate;
    iv. Case planning and Participant needs assessment;
    v. Medication administration assistance;
    vi. Wellness checks, security measures, meals and other services;

D. **Not Subject to Residential Tenancy Act:**
The Residential Tenancy Act (or successor legislation) (the 'Act) does not apply to this Agreement. The act does not apply to the accommodation because the accommodation is temporary accommodation made available in the course of providing the Participant with rehabilitative or therapeutic treatment services.

E. **Pets:**
All pets must be registered and are subject to approval by the Manager.
The Participant agrees to comply with the Participant Pet Ownership Agreement established or amended by XXXXX from time to time.

F. **Guests:**
    I. The building is only for the designated Participants, and only Participants may live in the building. If it is believed that a guest or other unauthorized person is living in or occupying a Participant unit, XXXXX reserves the right to ask the person to leave.
    II. The Participant may, subject to The Community's guidelines, permit an overnight guest in the unit for a maximum of ___________ nights out of a 365-day calendar year.
    III. No guests will be provided access to the building if they are not greeted by the Participant they are visiting in the lobby of the building. Guests must sign-in upon entry and show valid identification.
    IV. Guests must sign-out and leave through the front entrance.
    V. The agency reserves the right to refuse access to the building, to any person if it is determined the safety or security of participants or staff to be at risk or if the person

**Figure A1.** BC Housing's "generic" program agreement template, p. 2, retrieved 2023.

**Appendix B**

## PARTICIPATION IN PROGRAM

2.1    You agree to cooperate with PHS staff and to take part in the Program in good faith.

2.2    You agree that the Program is temporary with the possibility of becoming permanant, and is designed to help you succeed.  If you are not following the rules of the Program (including those in this Agreement), PHS can ask you to you leave the Accommodation.

2.3    The purpose of the Program is for Family Reunification and/or Family Stabilization. PHS understands that the process of Reunification can be long and difficult, so the program can last up to two years with possibility of extension, or an offer of permanant housing if available and appropriate.

2.4    You agree to work alongside PHS staff to accomplish the goals as are outlined upon intake with PHS Management.

2.5    You agree to disclose any outside organizational involvement (social workers, mental health teams, lawyers, ect.) so that we can support and collaborate around agreed upon information sharing and appointments where appropriate.

2.6    You agree to collaborating with PHS, and any family services agencies involved, to support in Reunification and/or Stabilization.

2.7    You agree to following the guidelines and expectations of any Family Service agencies that are involved as laid out by them.

2.8    You agree to be actively working on Reunification and/or Stabilization in all capacities, as required as part of your housing intake.

2.9    You agree to establishing a healthy family dynamic and respecting the dynamics of other families living at Station Street. You also agree to report any safety concerns in regards to children in the building and the building overall.

2.10    If applicable and/or appropriate, you agree to working with the PHS Indigenous Culture Support Worker and/or the PHS Social Worker around goals.

2.11    If applicable and/or appropriate you agree to using the PHS imedded Nurses for medication dispensing and our associated pharmancy, Community Apothicary. Station Street has an imbedded clinic and PHS nurses are able to administer medications and collaborate with other health care professionals.

**Figure A2.** *Cont.*

**PROGRAM FEE**

2.12  You agree that this Agreement is between you and PHS, and that other providers may be included as needed to support the needs of Family Reunification or Stabilization. The Program is funded by BC Housing, a crown agent, but BC Housing has no obligations to you under this Agreement.

2.13  If you receive Income Assistance from the Province of BC you will direct the Ministry of Social Development and Poverty Reduction to pay a fixed amount directly to PHS as your contribution to the costs of the Program (the "Program Fee"). In that case the Program Fee will be no higher than the maximum shelter portion of your income.

2.14  You must pay the Program Fee in advance, no later than the first day of each month.

2.15  Utilities, including basic cable and phone, are included in the Program Fee.

2.16  The Program Fee may change if your income or assets change.

**INCOME/PRIVACY**

2.17  You confirm that your income and assets amount to less than the BCH Housing Income Limit determined for this accommodation.

2.18  You agree that PHS can share your personal information with BC Housing, and BC Housing can verify that personal information to calculate the Program Fee.

**RIGHT TO OCCUPY**

2.19  You are the only person allowed to occupy the Accommodation. You will not allow anyone else to stay in it without expressed approval from management.

2.20  You agree that your right to occupy the Accommodation is temporary. When this Agreement ends, you will move out of the Accommodation.

2.21  The *Residential Tenancy Act* (or successor legislation) (the "Act") does not apply to this Agreement. The Act does not apply to the Accommodation because the Accommodation is a temporary accommodation made available in the course of providing you with rehabilitative or therapeutic treatment or services.

2.22  Any approved non-custodial guests can not exceed 8 days in one calendar month.

2.23  Only one approved guest at a time.

**COMMUNITY LIFE**

2.24  You agree to follow the rules PHS has established to help create a positive space. This means respecting the neighbourhood and:

- Not making too much noise at any time and especially in the nighttime and early morning hours.

**Figure A2.** Excerpt from 2023 Portland Hotel Society "program agreement.

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
