# Peer review of "Situating the Nonprofit Industrial Complex"

_socsci, doi:10.3390/socsci12100549_

Round 1
Reviewer 1 Report
Review of Situating the Nonprofit Industrial Complex
I think this is an excellent paper that makes an important argument about the limits of, and indeed the harm produced by, the nonprofit industrial complex (NPIC) in Vancouver, Canada. The article is well researched with extensive citations, and it is also well organized with clearly stated arguments and analysis. I have a few comments with the intention of improving an already strong paper.
The biggest suggest I have is for the authors to be even clearer about the exact contribution they are making, beyond the many critiques of non-profits, NGOs, NGOization that already exist (for an additional resource on the latter concept by Canadian-based scholars, which is not currently cited, see Choudry and Kapoor NGOization). The argument of the overall paper is clearly stated in the abstract: the “NPIC allows government and the wealthy to shirk responsibility for deepening health and social inequities, while shaping nonprofits’ revenue-generating objectives and weakening their accountability to community.” This is great. But again, what part of this argument is new versus arguments that others have made which these authors are reinforcing here? On page 2 it seems like the novelty of the article is that it expands the analysis of the NPIC to community policing and research centers. If this is indeed the main contribution, I would suggest making this clear in the abstract and the conclusions about why adding community policing and research centers to the NPIC critique is important. It does indeed seem like a novel contribution and important! However, my sense is that the authors probably have something broader and new to say about the NPIC in general, in addition to the great point (which again, could be emphasized more) about adding community policing and research centers to the concept of the NPIC.
A second suggestion that would strengthen the article… is there anything specific about the NPIC to Canada and/or Vancouver? The article is beautifully place-based, discussing the NPIC not only in a city but in a particular neighborhood and drawing on examples, etc., from that location. However, in the end, the critique of the NPIC seems generic to any country and any city. Is there anything specific about Vancouver or Canada that the authors could add to this analysis? For example, is the fact that Canada tends to have more public services than its neighboring United States mean something different for the deleterious impact of the NPIC? Having an argument that is both general about NPICs and then one that is specific about the NPIC in Canada would strengthen the contributions of the article.
Third, I think the author needs at least a short paragraph or section on methods, to explain how they collected the data they are using in the article. The article appears to be a summary/ /systematization of the literature on Nonprofits in Vancouver. This is fine, and I think this synthesizing of the literature does make a genuine contribution. However, this methodology needs to be stated more clearly. Conversely, were any of these examples offered in the article based on new data collection? My sense is no, but again, a short methods section would clarify these questions.
Lastly, how has philanthropy, both traditional and more strategic philanthropy shaped the NPIC in Vancouver, if at all? Megan Tompkins-Stange (2016) who writes about education reform shows how big foundations like Ford, Kellogg, Broad, Gates, are huge players in funding non-profits, however, Broad and Gates are also involved in a new form of aggressive policy intervention in education. Also, Kathryn Moeller in her book the Gender Effect shows how Nike Foundation shapes the programing of small non-profit organizations in Brazil. The authors of this article do mention funding sources as important, so I am curious if and how philanthropic foundations and the more “strategic” kinds of philanthropy that Tompkins-Stange describes are shaping and reshaping the NPIC in Vancouver.
A few smaller points. The authors point to the impact of the NPIC on Indigenous and racialized communities several times in the article, but the reader never gets a sense of the racialized context of Vancouver and the particular neighborhood they are examining. For example, in the discussion of housing and public housing policies, who are the folks living in these communities? Black Canadians, African immigrants, Latinx Canadians or Latinx immigrants, or none of the above? Again, the article attempts to take a place-based approach to this analysis, so I was surprised the authors were not more specific about the racial dynamics and patterns of immigration, etc., in the city.
Another smaller point, do the authors think it is possible for social movements that are fighting for structural changes in their cities to utilize the NPIC or the legal framework of nonprofits to support their more radical goals? Or, would the authors argue that even if this happens in very specific cases, the overall harm of the NPIC is the same?
Finally, on p. 12 the authors seem to turn their critique to the State itself, suggesting that non-state-funded and non-state organizational forms are the best path forward. This seems like a huge point, since most of the article I thought the authors were advocating for the state to take on these services again instead of devolving state responsibility to non-profits. In this final paragraph, it would be helpful for the authors to clarify what they think the role of the state should be, based on their analysis, and the relationship between the state and the more promising “models” listed on page 12.
Again, great paper, and these comments are just additional ideas to think about.
Author Response
Thank you very much for reviewing our article. It is deeply appreciated. Please see the attached document for full remarks.

Reviewer 2 Report
This manuscript critically examined nonprofit human service organizations in British Columbia. The authors seek to challenge the belief that nonprofits are inherently doing good, an important critique that has been made by many other scholars over time. The contribution of this manuscript is its analysis of Vancouver's nonprofit sector and how it is perpetuating inequity and preventing transformation of inequitable systems. With that said, I have some suggestions for improving the manuscript.
1. Painting nonprofits with such a broad brush diminishes the power of the analysis. Nonprofits, even the ones that receive funding and/or provide services, are complicated sites where inequities are reproduced and challenged. Some recognition of this would be important to note.
2. Some sentences are the opposite of what I think is intended by the authors. For example, on pg 9, 'fail to obscure the reality'. I think what the authors mean is 'obscure the reality'. pg. 10 'to scare monger youth..' Several other sentences are unclear.
3. Clearly describing the evidence used in the analysis early in the article would help. The authors state in the introduction that they are drawing a number of interconnected themes, but I wasn't clear what that meant. Themes based on the analysis of documents? news stories? The authors state that they offer a novel analysis of community policing but that was not clear to me. They indicate the number of nonprofits receiving funding from the Dept of Corrections and what the nonprofits do, but I was not sure how this was novel.
4. Clearly stating the literature that the article is contributing to would help the reader too. For example, there is an entire literature on nonprofit accountability and it seems to be the central concern of the article--whose interests are being served by these organizations--but this literature isn't cited
5. The authors make early statements, that nonprofits help manage dissent. This is an old argument, and important, but the evidence showing that they do this and how is missing.
NA
Author Response

(The authors gave the same response as above.)
